# Perioperative GABA Blood Concentrations in Infants with Cyanotic and Non-Cyanotic Congenital Heart Diseases

**DOI:** 10.3390/diagnostics11071149

**Published:** 2021-06-24

**Authors:** Angela Satriano, Alessandro Varrica, Alessandro Frigiola, Alessandro Graziosi, Caterina Di Battista, Adele Patrizia Primavera, Giacomo Centini, Antonio Maconi, Chiara Strozzi, Antonio D. W. Gavilanes, Luc J. Zimmermann, Hans J. S. Vles, Diego Gazzolo

**Affiliations:** 1Department of Pediatric Cardiac Surgery IRCCS San Donato Milanese Hospital, 20097 San Donato Milanese, Italy; a.varrica@virgilio.it (A.V.); alessandro.frigiola@grupposandonato.it (A.F.); 2Neonatal Intensive Care Unit, Department of Pediatrics, University of Chieti, 65100 Chieti, Italy; a.graziosi@hotmail.it (A.G.); caterina_di_battista@libero.it (C.D.B.); adeleprimavera@gmail.com (A.P.P.); dgazzolo@hotmail.com (D.G.); 3Department of Maternal Fetal and Neonatal Medicine, C. Arrigo Children’s Hospital Alessandria, 15100 Alessandria, Italy; giacomo.centini@ospedale.al.it (G.C.); amaconi@ospedale.al.it (A.M.); chiara.strozzi@libero.it (C.S.); 4Department of Pediatrics, Neonatology and Child Neurology, Maastricht University, 6200 MD Maastricht, The Netherlands; danilo.gavilanes@mumc.nl (A.D.W.G.); luc.zimmermann@mumc.nl (L.J.Z.); jsh.vles@mumc.nl (H.J.S.V.)

**Keywords:** GABA, brain, cardiopulmonary by-pass, newborn, children, reoxygenation, cooling

## Abstract

Perioperative stress detection in children with congenital heart disease (CHD), particularly in the brain, is still limited. Among biomarkers, γ-amino-aminobutyric acid (GABA) assessment in biological fluids appears to be promising for its regulatory action on the cardiovascular and cerebral systems. We aimed to investigate cyanotic (C) or non-cyanotic (N) CHD children for GABA blood level changes in the perioperative period. We conducted an observational study in 68 CHD infants (C: *n* = 33; N: *n* = 35) who underwent perioperative clinical, standard laboratory and monitoring parameter recordings and GABA assessment. Blood samples were drawn at five predetermined time-points before, during and after surgery. No significant perioperative differences were observed between groups in clinical and laboratory parameters. In C, perioperative GABA levels were significantly lower than N. Arterial oxygen saturation and blood concentration significantly differed between C and N children and correlated at cardiopulmonary by-pass (CPB) time-point with GABA levels. The present data showing higher hypoxia/hyperoxia-mediated GABA concentrations in C children suggest that they are more prone to perioperative cardiovascular and brain stress/damage. The findings suggest the usefulness of further investigations to detect the “optimal” oxygen concentration target in order to avoid the side effects associated with re-oxygenation during CPB.

## 1. Introduction

Congenital heart disease (CHD) is one of the most common types of birth defects, accounting for about 0.8% of full-term infants. Bearing in mind that CHD can occur 10 times more frequently in stillborn and premature than in term infants, the incidence of CHD must be considered significantly higher [1]. Notably, 90% of neonates with CHD survive into adulthood and more adults are living with CHD than children born with CHD [1]. The explanation resides in advances in open heart surgery, in anesthetic techniques and in cardiopulmonary by-pass (CPB) management that has substantially decreased the in-hospital mortality rate [2,3]. This expands the horizon to address functional neurologic and cardiac outcomes in long-term survivors [1,2,3]. In fact, CPB still remains a risky interventional procedure, characterized by a period of planned and deliberate hypoxia–ischemia (HI) injury, which is the price to pay in the treatment or palliation of CHD [4,5]. Hemodynamic and thermal changes occurring during CPB are well known to trigger a cascade of events such as ischemia–reperfusion injury, endothelial dysfunction, activation of complement, coagulation and inflammatory processes, leading to tissue and multi organ damage [3,6]. More recently, in healthy and CHD infants with complicated HI there is an exaggerated release of cytotoxic components, lasting from weeks to months, which characterizes the so-called third phase [7,8].

To date, the possibility of detecting perioperative stress, particularly of the brain, is still limited since clinical, laboratory and standard monitoring procedures may still be silent or unreliable [6,9]. Thus, a practical and sensitive marker, which is able to offer physicians useful information in clinical daily practice, is eagerly awaited. In this regard, it has recently been reported that cerebral excitatory amino acids can play an important role in the pathogenesis of ischemic brain injury [10].

Among biomarkers currently investigated in the perioperative period, Ɣ-aminobutyric acid (GABA) assessment in biological fluids appears to be promising [11]. GABA as an important neurotransmitter has been shown to regulate several neural signal pathways and physiological functions [11,12]. Results in animal models and further studies on humans provide evidence that GABA plays a regulatory role on cardiovascular and brain systems [12,13]. In the perioperative period, GABA has been shown to be involved in various biological activities such as anti-hypertension by vasopressin release modulation and analgesia through a nitric oxide mediated mechanism [13,14]. Data, however, on GABA perioperative monitoring in CHD infants is still lacking.

Therefore, in the present study we aimed to investigate, in infants who were complicated by cyanotic CHD (C) or non-cyanotic CHD (N), whether GABA blood levels in the perioperative period: (i) changed between C and N infants and (ii) correlated with standard perioperative parameters [3].

## 2. Materials and Methods

We conducted an observational study at our third level referral center for pediatric cardiac diseases in 68 CHD infants without pre-existing neurological disorders or other co-morbidities. CHD infants were corrected in C and N groups.

C group included: tetralogy of Fallot (*n* = 14), great arteries transposition (*n* = 6), tricuspid atresia (*n* = 6), truncus arteriosus (*n* = 1), double outlet right ventricle (*n* = 5) and single ventricle pathology (*n* = 1).

N group included: ventricular septal defect (*n* = 15), atrial septal defect (*n* = 8), partial anomalous pulmonary venous return and atrial septal defect (*n* = 6) and complete atrioventricular canal defects (*n* = 6) (Table 1).

Informed and signed consent from parents was obtained before patient inclusion in the study. This had been approved by the local human investigation committee.

All infants underwent clinical, standard laboratory and monitoring parameter recordings and GABA assessment upon admission to our unit. Blood samples were drawn at five predetermined times before, during and after surgery. In detail: before the surgical procedure (time 0, T0); during the surgical procedure after sternotomy before CPB (time 1, T1); at the end of CPB (time 2, T2); at the end of the surgical procedure (time 3, T3) and 24-h after the surgical procedure (time 4, T4).

### 2.1. Monitoring Parameters 

Clinical peripheral temperature, nasopharyngeal temperature, pump flow rate, mean arterial systolic and diastolic blood pressure (BP), left and right atrium BP (LA, RA), pulsed arterial oxygen tension (SaO_2_) and laboratory parameters (arterial blood pH and oxygen and carbon dioxide (PaCO_2_) partial oxygen pressure (PaO_2_), bicarbonate (HCO_3_), base excess (BE), hemoglobin concentration (Hb), hematocrit rate percentage (Ht) and glycaemia) were recorded at all sampling time-points.

### 2.2. Perioperative Near Infrared Spectroscopy (NIRS)

NIRS monitoring hemodynamic and oxygenation changes in the cerebral district were monitored in the peri-operative period, using a Sen Smart X-100 NIRS device (Nonin Medical, Plymouth, MN, USA). Self-adhesive transducers that contained light-emitting diodes and two distant Equanox Advance sensors (Nonin Medical) were fixed on the frontal lobe of the neonatal skull. Cerebral regional oxygen saturation (crSO_2_) was calculated by the in-built software. 

### 2.3. Anesthetic Technique

After premedication with Midazolam 0.5 mg/Kg body-weight (bw) (intramuscular), induction was achieved with oxygen and 3% Sevofluorane administered via mask (single breath induction), followed by intravenous Sufentanil 1 (g/Kg bw) and Vecuronium (0.15 mg/Kg bw). Maintenance was achieved with 3% Sevofluorane (except during CPB) and with additional doses of Sufentanil (0.5 g/Kg bw) and Vecuronium (0.1 mg/Kg bw) every 30–40 min. During CPB, in the absence of Sevofluorane, additional Midazolam at 0.2 mg/Kg bw dosage was given. Sufentanil infusion at 0.25 g/Kg bw was continued in the intensive care unit for sedation [15].

### 2.4. Cardiopulmonary by-Pass Management

CPB was established after systemic heparinization (3 mg/Kg bw) by standard single stage aortic and bicaval cannulation and maintained via non-pulsatile pump flow with a membrane oxygenator (Dideco Laboratories, Modena, Italy). Flow velocity was kept at 120–150 mL/Kg bw and mean arterial blood pressure at 45 mmHg; hypothermia was attained by core and surface cooling. Mean CPB duration time was 86 ± 27 min; mean rewarming time was 22 ± 9 min (mean ± SD), calculated from the final temperature during hypothermic circulatory arrest to 36.5 °C. The pump priming solution was composed of electrolyte solutions (Normosol-R 250 to 650 mL, Abbott Hospital Products, Abbott Park, IL, USA or Plasma-Lyte A, Travenol Laboratories, Inc., Deerfield, IL, USA), albumin (25%), heparin 1000 to 5000 units in the total solution, sodium bicarbonate (25–30 mEq/L) and packed red blood cells or frozen plasma. A standard circuit prime total volume was used according to bw as follows: bw < 4.5 Kg, 400 mL; bw > 4.5 Kg and bw < 7.5 kg, 600 mL; bw > 7.7 kg, 700 mL. Packed red blood cells (200 to 500 mL) were transfused as necessary to maintain a Ht level above 30% during CPB [16]. Protamine (1 mg for each mg of heparin) was administered at the end of CPB.

The α-stat regimen was used, and the PaCO_2_ maintained between 35 and 40 mmHg, without mathematical correction for the effects of the temperature, by varying the membrane oxygenator gas flow.

Modified ultrafiltration (MUF) was routinely performed before removal of arterial and venous cannulae. In the CPB circuit, the arterial line was connected to the inlet and the venous line to the outlet of the ultrafilter. As the patient was separated from the CPB, the clamp was removed from the inlet of the filter, allowing the blood to flow through the arterial line to the filter (10–15 mL/kg/min) and finally from the venous line to the RA. The filter allows the passage of molecules smaller than 65 kDa molecular weight. When it was necessary to maintain the intravascular volume and stabilize the hemodynamics, the blood returned via the venous reservoir and the venous cannula to the RA. This technique was performed until the Ht achieved the target of 35% [17].

### 2.5. GABA Measurement

Plasma samples (2 mL) were collected at the pre-determined monitoring time-points and the levels of GABA were measured using an ELISA kit (LDN Labor Diagnostika Nord, Nordhorn, Germany), as previously reported [18].

### 2.6. Neurological Follow-Up

Neurological development was assessed by physical examination, performed before surgery and at the 7th postoperative day, based on Amiel-Tison’s criteria [19]. In particular, resistance against passive movements, visual pursuit, reaching and grasping, and responses to visual and acoustic stimuli were tested by the same examiner, who did not know of the subjects’ pre-surgical condition.

### 2.7. Statistical Analysis

For the calculation of sample size, we used GABA concentration changes during CPB as the main parameter. As no basic data is available for CHD population, we assumed a decrease of 0.5 SD in GABA to be clinically significant. Indeed, considering an α = 0.05 and using a two-sided test, we estimated a power of 0.90 recruiting 30 C infants and 30 N as controls. We also added four cases to each group in order to avoid consent retirement, dropouts and early perioperative deaths. 

The Kolmogorov–Smirnov test showed values to have a Gaussian distribution, and the data was expressed as the mean and SD. Statistical significance was assessed using one-way ANOVA for repeated measures (followed by the post hoc Tukey test for multiple comparisons) and the unpaired *t*-test when only two groups were compared. Linear regression analysis was used for correlation between GABA, CPB and laboratory parameters. Statistical significance was set at *p* < 0.05.

## 3. Results

### 3.1. Perioperative Parameters 

In Table 1 the characteristics of the infants in the two studied groups are reported. No overt neurological injury was observed in surviving patients during the first week after surgery according to standard perioperative monitoring procedures (data not shown). There were no discernible clinical differences in the favorable outcomes of patients.

No significant differences were found in the two groups regarding gender, neurological examination and perioperative in-hospital mortality (*p* > 0.05, for all), whilst age and weight on admission to the study differed between groups (*p* < 0.05, for both). 

Intraoperative parameters such as CPB, aortic cross clamping, circulatory arrest and cooling durations were significantly higher (*p* < 0.05, for all) in the C group. The incidence of MUF did not differ (*p* > 0.05) between groups. 

In Table 2, clinical parameters recorded in the perioperative period of the two studied groups are reported. No significant statistical differences (*p* > 0.05, for all) were observed regarding laboratory parameters measured at the different monitoring time-points, such as Hb, Ht, pH, pCO_2_, HCO_3_, BE and glycaemia arterial blood levels. Moreover, no significant (*p* > 0.05, for all) differences were observed regarding monitoring parameters, such as HR, LA and RA BP, systolic and diastolic BP.

PaO_2_ pattern of concentration in N infants was characterized by a decline from T0 (before surgical procedure) to T1 (before CPB) when it started to increase, reaching its highest peak at the end of CPB (T2). It was from T3 (at the end CPB) onwards that PaO_2_ started to decrease, reaching its lowest point at 24 h from surgical procedure completion (T4).

In the C infants PaO_2_ pattern of concentration started to increase from T0, reaching its highest peak at the end of CPB (T2). Once more, from T3 onwards, PaO_2_ started to decrease, reaching its lowest point at T4.

PaO_2_ significantly differed in C infants at T0, T1 and T4 monitoring time-points, while no differences (*p* > 0.05, for all) were observed at T2-T4 time-points.

SaO_2_ pattern was in part superimposable on PaO_2_: significant (*p* < 0.05, for both) lower values in C at T0 and T1; no significant differences (*p* > 0.05, for all) at T2-T4 (Figure 1).

### 3.2. Near-Infrared Spectroscopy Recordings

NIRS perioperative pattern in the N group was characterized by a progressive decrease in crSO_2_ values from T0 (before surgery), reaching the lower dip (*p* < 0.001) at T3 (end of surgical procedure) and returning to baseline values at 24 h from surgery. 

In the C group, crSO_2_ pattern was characterized by a flat trend and no differences were found among T0-T4 time-points. 

Higher perioperative crSO_2_ values (*p* < 0.01, for all) were observed at all monitoring time-points in N children (Figure 2).

### 3.3. GABA Measurements

GABA levels were measurable in all samples collected at different monitoring time-points.

In N children, GABA blood concentrations showed a flat pattern from T0 to T4 (*p* > 0.05 for all) except for a significant (*p* < 0.05) increase at T1 (before CPB) (Figure 3).

In the C children, GABA blood concentrations showed a pattern characterized by a significant (*p* < 0.05, for all) progressive decrease in biomarker levels from T0 to T4 reaching the lowest point at T2.

When GABA levels were compared between the two studied groups, in the perioperative period, we found in C higher (*p* < 0.001) GABA levels at T0; no differences (*p* > 0.05) at T1; lower (*p* < 0.01) GABA levels at T2; no differences (*p* > 0.05) at T3; and lower (*p* < 0.01) GABA levels at T4.

Linear regression analysis showed a significant correlation between GABA and CPB parameters such as: CPB duration (R = −0.65; *p* < 0.001), cross clamping (R = −0.59; *p* < 0.001) and body core temperature (R = −0.67; *p* < 0.001). Moreover, at T2, GABA levels correlated with PaO_2_ (R = −0.73; *p* < 0.01) and SaO_2_ (R = −0.68; *p* < 0.01).

## 4. Discussion

There is growing evidence that despite recent technological advances, the early detection of perioperative brain stress/damage in CHD children still remains one of the major unsolved issues [1,4,6]. During open-heart surgery, especially in the CPB phase, front-line physicians are aware that children are facing harmful effects on all body organs through a hemodynamic temperature-dependent phenomenon [6,20]. The finding is of relevance in C children that have been shown to be particularly at risk of HI and reperfusion injuries [21]. Therefore, identification and validation of biochemical parameters for brain monitoring, in CPB-instrumented infants according to different CPB strategies, could contribute to the early detection of brain damage [5,14,21,22].

In the present study, we found that in children requiring open heart surgery for CHD correction, the perioperative blood concentrations of a brain-tissue function and damage, such as GABA, significantly changed during surgical procedure, particularly in C cases. Furthermore, GABA levels correlated with the length of different CPB phases (i.e., cooling, cross-clamping) and core body temperature. Notably, GABA levels also correlated with laboratory and monitoring parameters recorded at the end of CPB procedure, such as PaO_2_ and SaO_2_.

To our knowledge the finding of a significant difference in GABA levels between C and N children constitutes the first observation. In this light, GABA pattern of concentration in the two studied groups deserves further consideration. In particular, GABA levels in C children were higher before surgical procedure, superimposable on CHD before CPB and significantly lower from the end of CPB up to 24 h from surgery completion. Bearing in mind the main known GABA functions, to date controversial and still a matter of debate, the most intriguing one regards its role as an inhibitor or pro-activator factor under different stimuli [13,23,24]. This especially holds for hypoxia, which causes dysfunction of excitatory and inhibitory neurotransmission finally resulting in brain damage [24,25]. The fact is of relevance and offers additional support to the notion that C children complicated by in-utero chronic hypoxia are more prone to the so-called deliberate open-heart surgery mediated HI insult. The finding confirms previous observations suggesting that C children per se are more inclined to different CPB side-effects, such as cooling and rewarming as suggested by GABA correlation with CPB, cross clamping and core body temperature. This latter point particularly regards the re-oxygenation phase that is characterized by a sustained hyperoxia, thus exposing CHD children to oxidative stress damage [25]. Another explanation may reside in the therapeutic strategies to date performed in the perioperative management of open-heart surgery. This especially refers to administration of GABA agonists, such as progabide, diazepam, phenobarbital and propofol that increase inhibitory chloride conductance or up-regulate the effect of synaptic released GABA on the GABA-A receptors [23,24,25,26]. In this light, it is possible to argue that the increased GABA levels detected during surgical procedure, but before CPB, are related to sedative strategy performance. Conversely, the significant decrease in GABA levels during CPB procedure merits further consideration [27]. The main explanations reside in: (i) a hemodynamic and temperature-time-dependent mechanism, as shown by the correlation between GABA levels, and CPB and cooling and clamping durations, respectively, and (ii) a putative early detection of subclinical brain damage. In this regard, it is of relevance that no overt neurological injury was diagnosed during hospital stays and (iii) CPB strategy and its cooling phase known to affect patients’ metabolism, including hemodynamic parameters and kinetic therapeutic strategies [26,27]. In this regard, hypothermia can activate a cascade of events characterized by a delay in sedative, inotrope and neuroprotective strategy performance finally leading to GABA release inhibition up to 24 h after surgery [8]. Altogether, it is reasonable to argue that GABA changes in C infants who have undergone open heart surgery are mainly due to CPB procedure itself. Further multicenter studies in the wider population aimed at optimizing CPB management are needed [8,20,25].

In the present study we also found that arterial oxygen pulsed saturations and partial pressures were significantly lower in C than N children in the perioperative period. Data is corroborated by lower crSO_2_ values in C children at different monitoring time-points supporting the notion of a lower oxygenation status in the cerebral bloodstream of C children.

The pattern of no significant differences in PaO_2_ and SaO_2_ levels at the end of CPB procedure between C and N groups warrants further consideration. The finding is of relevance and suggestive of a relative hyperoxia status/damage in C children. In this regard, results in animal models and in humans showed: (i) a lower incidence of brain damage when weaned from CPB by means of a controlled normoxic re-oxygenation mode [27], (ii) increased lipid peroxidation and decreased antioxidant reserve capacity through endogenous radical scavengers [6,27,28,29,30,31,32] and (iii) increased oxygen toxicity thus affecting endothelial permeability [4,27]. Altogether, it is reasonable to argue that hyperoxia status/damage in C children is restricted to CPB phase as shown by lower PaO_2_ and SaO_2_ in C from the end of CPB up to 24 h from surgery [25]. Another explanation may reside in CPB-mediated blood-brain barrier permeability changes decreasing GABA release in systemic circulation [33]. In any case, further multicenter investigations in the wider population should be requested to elucidate GABA role in brain stress/damage in C children.

Lastly, we recognize that the present study has several limitations. In particular: (i) the evaluation of the impact of biomarkers on clinical daily practice is today still a topic of debate, especially in neonatal and pediatric populations. In this regard, international health care institutions (i.e., Food and Drugs Administration, European Medicine Agency, National Institution of Health) stated a series of clinical and laboratory criteria for the inclusion of neurobiomarkers in the daily clinical routine. The main one regards the validation by means of correlation with the “so called” standard of care procedures, such as cerebral magnetic resonance imaging (MRI) [10,34]. At this stage, there is no conclusive data on the correlation of biomarkers with MRI, as well as for other monitoring procedures (i.e., EEG, NIRS etc.), (ii) the small population recruited in the present series, (iii) the absence of biomarker correlation with long term neurological follow-up and (iv) the lack of reference curves for new neuro-biomarkers including GABA. Further investigations aimed at addressing the aforementioned issues are eagerly awaited.

## 5. Conclusions

In conclusion, the present data in C children undergoing open heart surgery and CPB suggests the need for careful and controlled systemic re-oxygenation. Results open the way to further investigations aimed at detecting the “optimal” PaO_2_ target [32] according to CHD complexity by a panel of neurobiomarkers among which GABA can play a relevant role.

## Figures and Tables

**Figure 1 diagnostics-11-01149-f001:**
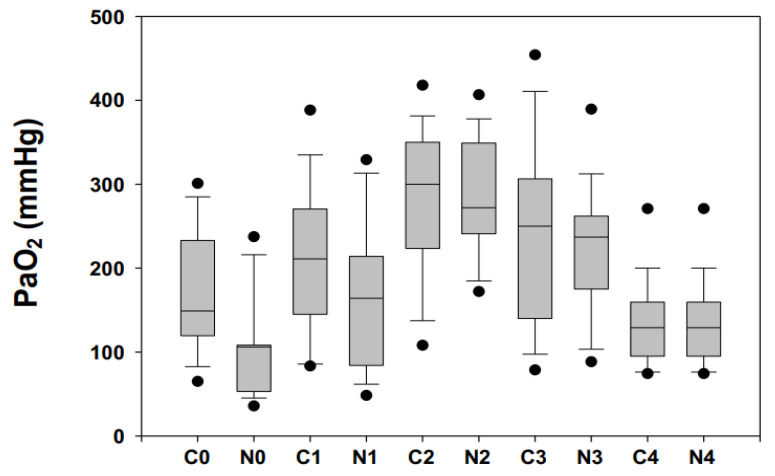
Arterial oxygen partial pressure levels (PaO_2_, mmHg) expressed as median and 5–95° centiles (•) ranges in cyanotic (C) and acyanotic (N) congenital heart disease children measured before the surgical procedure (T0); during the surgical procedure after sternotomy before cardiopulmonary bypass (CPB) (T1); at the end CPB (T2); at the end of the surgical procedure (T3); and at 24 h after the surgical procedure (T4). PaO_2_ significantly differed in C infants at T0, T1 and T4 monitoring time-points while no differences (*p* > 0.05, for all) were observed at T2-T4 time-points.

**Figure 2 diagnostics-11-01149-f002:**
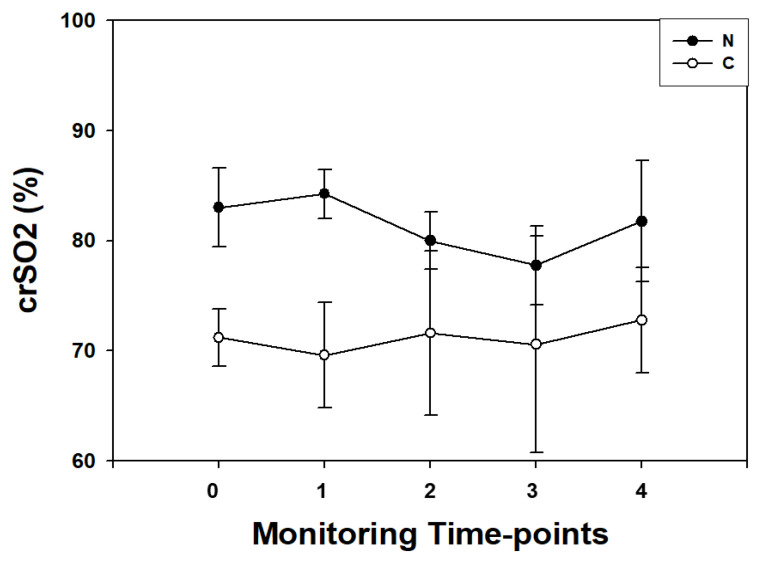
Cerebral regional oxygenation status (crSO_2_) expressed as median and 5–95° centiles ranges in cyanotic (C; •) and non-cyanotic (N; o) congenital heart disease children measured before the surgical procedure (T0); during the surgical procedure after sternotomy before CPB (T1); at the end CPB (T2); at the end of the surgical procedure (T3); and at 24 h after the surgical procedure (T4). crSO_2_ was significantly lower (*p* < 0.01, for all) in C infants at all monitoring time-points.

**Figure 3 diagnostics-11-01149-f003:**
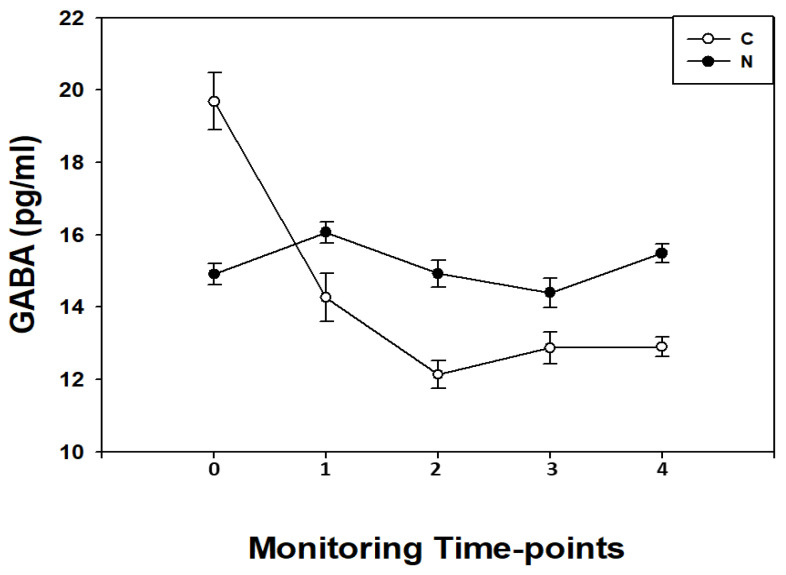
GABA blood concentrations (pg/mL) expressed as median and interquartile ranges in cyanotic (C; o) and non-cyanotic (N; •) congenital heart disease children measured before the surgical procedure (T0); during the surgical procedure after sternotomy before CPB (T1); at the end CPB (T2); at the end of the surgical procedure (T3), and at 24 h after the surgical procedure (T4). In C GABA levels when compared with N children were higher (*p* < 0.001) at T0; no differences (*p* > 0.05) at T1; lower (*p* < 0.01) at T2; and no differences (*p* > 0.05) at T3; lower (*p* < 0.01) at T4.

**Table 1 diagnostics-11-01149-t001:** General characteristics, laboratory parameters and main interventions in cyanotic (C) and non-cyanotic congenital heart disease (N) children admitted into the study. Data are expressed as means ± SD. * *p* < 0.05.

Parameters	C (*n* = 33)	N (*n* = 35)
Age (months)	23 ± 11	30 ± 8 *
Weight (Kg)	12 ± 1	16 ± 3 *
Gender (F/M)	11/23	12/22
Laboratory parameters		
Hemoglobin (g/dL)	13.5 ± 1.4	12.9 ± 2.0
Hematocrit (%)	39.9 ± 3.9	38.5 ± 4.9
Platelet count (10^3^/mmc)	329 ± 107	335 ± 99
Creatinine (mg/dL)	0.5 ± 0.3	0.5 ± 0.2
Urea (mg/dL)	39 ± 11	37 ± 12
LDH (UI/L)	565 ± 199	532 ± 206
CK (UI/L)	168 ± 112	173 ± 109
Neurological examination before surgery		
Normal/suspect/abnormal	34/0/0	34/0/0
Main interventions		
CPB (min)	88 ± 22	59 ± 25 *
Filtration (*n*/total)	19/34	17/34
Clamping (min)	79 ± 25 *	47 ± 29 *
Circulatory arrest (*n*/total)	8/34	1/34 *
Cooling (°C)	31 ± 3	27 ± 2 *

Abbreviations: LDH: lactatedehidrogenase; CK: creatinphosphokinase, CPB: cardiopulmonary bypass.

**Table 2 diagnostics-11-01149-t002:** Laboratory and monitoring parameters in cyanotic (C) and non-cyanotic congenital heart disease (N) children at different monitoring time-points: before the surgical procedure (T0); during the surgical procedure after sternotomy before CPB (T1); at the end of the cross-clamp CPB phase (T2); at the end of CPB (T3); at the end of the surgical procedure (T4); and, at 24 h after the surgical procedure (T5). * *p* < 0.05.

Parameters	T0 N	T0 C	T1 N	T1 C	T2 N	T2 C	T3 N	T3 C	T4 N	T4 C
Hb (g/dL)	12.5 ± 2.1	12.2 ± 1.9	11.8 ± 2.3	12.1 ± 2.2	11.5 ± 2.9	11.7 ± 1.9	11.4 ± 1.8	11.3 ± 2.6	11.8 ± 1.8	11.6 ± 1.4
Ht (%)	35.7 ± 2.8	35.2 ± 2.2	34.1 ± 4.9	34.9 ± 3.7	32.9 ± 7.3	33.5 ± 3.1	32.8 ± 3.7	33.6 ± 2.2	34.7 ± 3.7	34.3 ± 4.1
pH	7.31 ± 0.10	7.34 ± 0.12	7.36 ± 0.11	7.35 ± 0.12	7.38 ± 0.09	7.36 ± 0.10	7.37 ± 0.11	7.36 ± 0.13	7.41 ± 0.08	7.40 ± 0.09
PaCO_2_ (mmHg)	35.4 ± 4.9	35.6 ± 5.3	35.4 ± 3.8	36.2 ± 4.5	34.7 ± 5.2	36.1 ± 3.1	36.8 ± 4.4	35.5 ± 3.4	36.3 ± 5.7	35.8 ± 6.0
PaO_2_ (mmHg)	205.3 ± 94	107 ± 18 *	186 ± 72	150 ± 48 *	266 ± 68	290 ± 67	185 ± 78	138 ± 63 *	183 ± 55	138 ± 46
H_2_CO_3_	23.1 ± 3.1	22.5 ± 2.3	21.8 ± 2.9	21.9 ± 4.6	21.5 ± 3.8	21.9 ± 4.0	22.7 ± 1.5	21.8 ± 2.3	22.4 ± 1.7	22.1 ± 1.5
BE	0.3 ± 2.4	0.2 ± 1.6	−1.9 ± 3.3	−1.5 ± 3.1	−3.0 ± 2.3	−2.3 ± 2.9	−0.3 ± 0.9	0.8 ± 2.1	1.6 ± 1.7	1.5 ± 1.4
SaO_2_ (mmHg)	98.4 ± 2.6	82.4 ± 7.8 *	97.8 ± 0.4	84.2 ± 8.7 *	97.5 ± 1.9	96.0 ± 2.1	95.9 ± 1.6	94.8 ± 3.8	95.5 ± 4.6	93.1 ± 4.7
HR (bpm)	106 ± 10	111 ± 15	112 ± 13	111 ± 10	124 ± 12	118 ± 15	124 ± 14	134 ± 16	127 ± 18	123 ± 13
LA BP (mmHg)	9.0 ± 3.7	7.9 ± 2.9	7.7 ± 3.0	7.4 ± 3.9	9.3 ± 3.5	8.9 ± 4.3	9.3 ± 5.1	8.6 ± 3.2	9.4 ± 4.1	9.3 ± 3.9
RA BP (mmHg)	9.4 ± 2.5	9.2 ± 1.8	8.8 ± 1.4	8.9 ± 1.6	8.8 ± 2.3	8.4 ± 2.8	8.9 ± 2.7	7.8 ± 2.2	10 ± 2.9	11 ± 2.3
S BP (mmHg)	89 ± 12	86 ± 13	87 ± 19	86 ± 15	87 ± 14	85 ± 12	88 ± 16	86 ± 13	97 ± 10	95 ± 15
D BP (mmHg)	43 ± 14	44 ± 9	54 ± 11	53 ± 13	55 ± 10	56 ± 9	50 ± 11	54 ± 14	58 ± 12	56 ± 10
Glycaemia (mg/dL)	103 ± 10	105 ± 8	117 ± 10	115 ± 14	129 ± 16	117 ± 10	110 ± 22	124 ± 13	118 ± 15	120 ± 11

Abbreviations: Hemoglobin (Hb); Hematocrit (Ht); carbon dioxide (PaCO_2_) and oxygen (PaO_2_) partial pressures; bicarbonate (HCO_3_); base excess (BE); pulsed arterial oxygen saturation (SaO_2_); Heart rate (HR); Left atrium blood pressure (LA BP); Right atrium blood pressure (RA BP); Systolic blood pressure (S BP); Diastolic blood pressure (D BP).

## Data Availability

IRCCS San Donato Milanese Hospital, San Donato Milanese, Italy.

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
