# Peer review of "Perioperative GABA Blood Concentrations in Infants with Cyanotic and Non-Cyanotic Congenital Heart Diseases"

_diagnostics, 2021, doi:10.3390/diagnostics11071149_

Round 1
Reviewer 1 Report
Satriano et al. presented revised manuscript investigating peripheral blood GABA levels as a potential biomarker of children undergoing CPB. Authors now present NIRS data, however, authors failed to address all the other critiques raised by this reviewer:
- There is no normalization of the data to weight, age, procedure duration.
- Authors did not discuss the possibility of hypooxygenemia being the cause of GABA difference in this cohort.
- Again, no reference ranges for GABA are mentioned, therefore it is unclear whether GABA can possibly be used as a clinical biomarker.
- NIRS data presented are within normal range. Moreover, authors did not show any statistics in the Figure 2. Was there a significant differences between experimental groups?
Author Response
Satriano et al. presented revised manuscript investigating peripheral blood GABA levels as a
potential biomarker of children undergoing CPB. Authors now present NIRS data, however, authors
failed to address all the other critiques raised by this reviewer:
1. There is no normalization of the data to weight, age, procedure duration.
We already argued with reviewer on the point and unfortunately, we are not able to fulfill his/her requirement since it was not the aim of the study.
2. Authors did not discuss the possibility of hypooxygenemia being the cause of GABA difference in this cohort.
Discussion section has been expanded as suggested. We trust that now is clearer.
3. Again, no reference ranges for GABA are mentioned, therefore it is unclear whether GABA can possibly be used as a clinical biomarker.
We already argued with reviewer on the point and unfortunately, we are not able to fulfill his/her requirement since it was not the aim of the study.
4. NIRS data presented are within normal range. Moreover, authors did not show any statistics in the Figure 2. Was there a significant differences between experimental groups?
In the previous MS revision version, in order to avoid Figure 2 redundant symbols and data, we provided in the dedicated legend a detailed description of statistical differences between C and N groups. Anyway, statical differences were also reported in the results section. We trust that now is
clearer.
Reviewer 2 Report
The manuscript is improved and the results interesting.
I am not an expert of the English language but there are still grammatical and spelling mistakes and improvements to be made to gain more clarity in the text.
Author Response
The manuscript is improved and the results interesting. I am not an expert of the English language but there are still grammatical and spelling mistakes and improvements to be made to gain more clarity in the text.
We want to thank reviewer for kind comments and useful suggestion. MS has been revised by our Institution native english speaker and typing errors have been avoided from all the text. Results have been improved as requested. We trust that now is clearer.
Reviewer 3 Report
Satriano and colleagues presented an interesting paper “GABA Blood Concentrations in Children Subjected to Cardio-pulmonary by-pass”
Overall, the paper interesting and offers an original evaluation of GABA levels in the perioperative period in patients with congenital heart disease. The authors aim to evaluate “whether GABA blood levels in the peri-operative period: i) changed between CHDc and CHD infants, ii) correlated with standard perioperative parameters”. Since during the paper a continuous comparison between cyanotic and non-cyanotic patients is made, I suggest adding this in the title. I also suggest a comparison of GABA level according to the complexity of the CHD.
Is there any correlation between GABA level and outcome?
Line 63: I suggest avoiding simple CHD and use cyanotic and non-cyanotic CHD since as specified in lines 70-72 noncyanotic CHD included complex CHD such as DORV and AVSD. I suggest using a uniform definition ot the two groups: cyanotic (C) and acyanotic (N) is sometimes used and, in figure A3 cyanotic and control is used.
Line 68 “without pre-existing neurological disorders or other co-morbidities”- The pre-existing neurological disorders are assessed clinically or with EEG/imaging Echo or MRI. I suggest adding the results of neurological evaluation. I suppose that evaluation is very heterogeneous since the age is varies from 23 ± 11 months and 30 ± 8 months. How did you define comorbidities? No syndromic patients were included (there was no patient with Down Syndrome even if you included 6 complete AVSD)
Line 72 Change! double out-let right ventricle” with Double outlet right ventricle
Table 1A the last line has different characters
Figure A 3 change contrrol with control
You say that perioperative mortality is the same in both groups, but you do not specify the in-hospital mortality.
Author Response
Satriano and colleagues presented an interesting paper “GABA Blood Concentrations in Children Subjected to Cardio-pulmonary by-pass”. Overall, the paper interesting and offers an original evaluation of GABA levels in the perioperative period in patients with congenital heart disease. The authors aim to evaluate “whether GABA blood levels in the peri-operative period: i) changed between CHDc and CHD infants, ii) correlated with standard perioperative parameters”.
• Since during the paper a continuous comparison between cyanotic and non-cyanotic patients is made, I suggest adding this in the title.
We want to thank reviewer for kind comments and useful suggestion. MS title has been changed as requested.
• I also suggest a comparison of GABA level according to the complexity of the CHD.
Unfortunately, we were not able to address the reviewer’s request. The explanation resides in the small study population that did not allow to reach a robust result. In fact, we needed a wider population in order to exclude peri-operative bias by means of a multivariable analysis. Further studies are
in this respect justified.
• Is there any correlation between GABA level and outcome?
No overt neurological injury at short-term follow-up has been found.
• Line 63: I suggest avoiding simple CHD and use cyanotic and non-cyanotic CHD since as specified in lines 70-72 noncyanotic CHD included complex CHD such as DORV and AVSD. I suggest using a uniform definition ot the two groups: cyanotic (C) and acyanotic (N) is sometimes used and, in figure A3 cyanotic and control is used.
According to reviewer’s suggestion we changed group definition. We trust that now is clearer.
• Line 68 “without pre-existing neurological disorders or other co-morbidities”- The pre-existing neurological disorders are assessed clinically or with EEG/imaging Echo or MRI. I suggest adding the results of neurological evaluation. I suppose that evaluation is very heterogeneous since the
age is varies from 23 ± 11 months and 30 ± 8 months. How did you define comorbidities? No syndromic patients were included (there was no patient with Down Syndrome even if you included 6 complete AVSD)
Results section has been expanded as requested. However, as shown in Table 1 ultrasound imaging recording was not possible due to patients age (> 12 months) and due to infrastructure limitation perioperative continuos aEEG was not performed. MRI was not performed in absence of any clinical
and laboratory pattern suggestive of brain distress/damage.
• Line 72 Change! double out-let right ventricle” with Double outlet right ventricle
Typing errors have been avoided.• Table 1A the last line has different characters
Typing errors have been avoided.
• Figure A 3 change contrrol with control
Typing errors have been avoided.
• You say that perioperative mortality is the same in both groups, but you do not specify the inhospital mortality.
Introduction has been changed as requested. We trust that now is clearer
Reviewer 4 Report
I can only comment on the GABA part of the study.
Any idea of the source of GABA in the blood?
Any male/female differences observed?
Care to comment on the effects of the anesthetics known effects on GABA receptors?
The authors might like to refer to this very recent paper: Plasma gamma-aminobutyric acid (GABA) levels and posttraumatic stress disorder symptoms in trauma-exposed women: a preliminary report, Hall et al., Psychopharmacology (2021) 238:1541–1552 https://doi.org/10.1007/s00213-021-05785-z
Author Response
We want to thank reviewer for kind comments and useful suggestion.
• Any idea of the source of GABA in the blood?
Literature data reports a GABA ubiquitary site of releasing. The issue has been stated in the discussion section.
• Any male/female differences observed?
Unfortunately, we were not able to observe any gender differences due to small sample size population.
• Care to comment on the effects of the anesthetics known effects on GABA receptors? The authors might like to refer to this very recent paper: Plasma gamma-aminobutyric acid (GABA) levels and posttraumatic stress disorder symptoms in trauma-exposed women: a preliminary report, Hall et
al., Psychopharmacology (2021) 238:1541–1552 https://doi.org/10.1007/s00213-021-05785-z
Reference list has been updated as suggested.
Reviewer 5 Report
I had a pleasure to read this interesting study that address vital aspects of surgery in infants that require cardiopulmonary bypass. This puts child at increased risk of cognitive deterioration and brain injury. One way of protection against brain injury is shortening time of CPB use. Authors presented very valuable idea of GABA and re oxygenation. Study is well written and I believe that it merits further investigations
Author Response
I had a pleasure to read this interesting study that address vital aspects of surgery in infants that require cardiopulmonary bypass. This puts child at increased risk of cognitive deterioration and brain injury. One way of protection against brain injury is shortening time of CPB use. Authors presented
very valuable idea of GABA and re oxygenation. Study is well written and I believe that it merits further investigations.
We want to thank reviewer for kind comments and useful suggestion.
Round 2
Reviewer 1 Report
1. Authors state that they already argued with reviewer on the point that there is no normalization of the data, and state "we are not able to fulfill his/her requirement since it was not the aim of the study." Data normalization for the confounders is an integral part of any study. Therefore stating that "it was not the aim of the study" rather implies that authors just gathered data ignoring the proper analysis.
2. Where in "Results" authors provide the statistics for the NIRS data that this reviewer requested? They only provided the p value. If you are making a claim that datapoints that are within the normal range are predictive of pathological condition/outcome, you have to present a proper regression analysis that demonstrates the trend although showing normal values.
Author Response
Reviewer 1
- Authors state that they already argued with reviewer on the point that there is no normalization of the data, and state "we are not able to fulfill his/her requirement since it was not the aim of the study." Data normalization for the confounders is an integral part of any study. Therefore, stating that "it was not the aim of the study" rather implies that authors just gathered data ignoring the proper analysis.
The aims of the present study were to compare GABA blood levels between cyanotic and non- cyanotic CHD children and its correlations with CPB different phases. We were not claiming for GABA reference curves gender or weight corrected. Literature data in this respect is lacking.
However, in order to fulfil reviewer’s request, based on a preliminary statical analysis in agreement with Wesley et al. (see for review), the sample size calculation would result in a cohort of about 80-100 cases per year. Therefore, we will need about 2000 cases. We trust that now the aims of our study are clearer.
- 2. Where in "Results" authors provide the statistics for the NIRS data that this reviewer requested? They only provided the p value. If you are making a claim that datapoints that are within the normal range are predictive of pathological condition/outcome, you have to present a proper regression analysis that demonstrates the trend although showing normal values.
In the results section we reported data on different cerebral oxygenation status in the two studied populations and their relative p–values. In Figure 1 longitudinal cerebral oxygenation results have been reported. We did not report NIRS data in the results section in order to avoid redundant data (text and figure). Anyway, we are available to include NIRS values and their interquartile ranges if needed.
We did not state that NIRS patterns were correlated with any pathological condition/outcome. The fact that all values were within normal ranges does not mean that there were no statistical differences between groups.
We trust that now the MS in the present revised form will meet the criteria for publication on Your Journal.
Best regards.
Angela Satriano, MD
Reviewer 3 Report
It was my pleasure to see the changes apported to the paper “Perioperative GABA Blood Concentrations in Infants With 2 Cyanotic and Non-Cyanotic Congenital Heart Diseases”
I suggest avoiding the term simple CHD since AVSD is not a simple CHD. "Therefore, in the present study we aimed to investigate, in infants who were complicated by cyanotic CHD (C) or simply CHD (N), whether GABA blood levels in the peri-operative period: i) changed between C and N infants, ii) correlated with standard peri-operative parameters [3]"
I suggest changing (10 15 mL/kg/min) change into (10 - 15 mL/kg/min).
The authors talk about surviving patients but there is no reference to the survival rate, is there any difference in the survival rate between groups? Are the GABA level associated with a poorer outcome?